ecology

penguin, camera, patch size, prey, foraging, predator–prey interactions

**Author for correspondence:**
G. J. Sutton
e-mail: g.sutton@deakin.edu.au

# Quantity over quality? Prey-field characteristics influence the foraging decisions of little penguins (*Eudyptula minor*)

## G. J. Sutton and J. P. Y. Arnould

School of Life and Environmental Sciences, Faculty of Science and Technology, Deakin University, 221 Burwood Highway, Burwood, VIC 3125, Australia

GJS, 0000-0002-9798-0281; JPYA, 0000-0003-1124-9330

Quantifying prey characteristics is important for understanding the foraging behaviour of predators, which ultimately influence the structure and function of entire ecosystems. However, information available on prey is often at magnitudes which cannot be used to infer the fine-scale behaviour of predators, especially so in marine environments where direct observation of predator–prey interactions is rarely possible. In the present study, animal-borne video data loggers were used to determine the influence of prey type and patch density on the foraging behaviour of the little penguin (*Eudyptula minor*), an important predator in southeastern Australia. We found that numerical density positively influenced time spent foraging at a patch. However, when accounting for calorific value in density estimates, individuals spent longer at dense patches of low-quality prey. This may reflect a trade-off between capture effort and calorific gain as lower quality prey were captured at higher rates. During the breeding season, foraging trip distance and duration is constrained by the need to return to the colony each day to feed offspring. The results of the study suggest that, under these spatio-temporal constraints, little penguins maximize foraging performance by concentrating efforts at larger quantities of prey, irrespective of their calorific quality.

## 1. Introduction

Successfully finding and capturing prey is the most important aspect of an animal's fitness as it provides the energy required to carry out important daily processes [1,2]. When prey availability changes, animals are expected to adapt their responses in order to continue maximizing their energy gain

while minimizing energy expenditure [3,4]. Energy acquisition through efficient foraging strategies directly impacts an animal's fitness, especially during important life-history periods such as breeding [5]. Understanding how foraging and energy use are intertwined, especially during periods of high-energy demand, is important for predicting population trends in a changing environment [6].

Predators have the potential to place extraordinary pressures on their prey populations, which can lead to changes in the structure of ecosystems [7,8]. The strength of top-down effects in any system is influenced by the population of predators and the availability of prey [9,10]. In terrestrial environments, researchers have defined and quantified the ecological roles of many large carnivore species [11–13]. However, in marine environments, with many predator species hunting at depth, direct assessment of foraging behaviour and predator–prey interactions is logistically difficult. While there have been attempts to quantify prey presence through the use of meso-scale environmental characteristics (e.g. sea surface temperature (SST) and Chlorophyll-*a*), and at-sea sonar surveys, studies have revealed mis-matches between these variables and the distribution of predators suggesting time lags associated with these data [14]. The use of telemetry such as GPS [15], dive recorders [16], accelerometers [17,18], stomach temperature and mandibular sensors [19–22] have also been used to infer prey capture attempts. However, such devices do not provide the ability to directly quantify an animal's prey-field (i.e. the type and density of prey which is immediately available to the forager), a critical component in quantifying predator–prey interactions and gaining further insight into ecosystem functioning [23,24].

In the marine environment, pelagic prey is distributed in discrete patches at varying spatio-temporal scales [25]. In order to observe predator–prey encounters, animal-borne video data loggers are useful in quantifying predator movements in response to their associated prey-fields [23,24,26,27]. However, few studies have assessed foraging behaviour in response to prey abundance and quality (i.e. the energetic value) [28–30]. It is these direct prey-field measurements that allow insight into the foraging decisions of predators and how behaviour may adapt to changes in prey availability because of natural and anthropogenic environmental variation.

Breeding seabirds are ideal species to study predator–prey interactions as it is possible to obtain large amounts of information on the foraging behaviour of many individuals due to their regular nest attendance patterns and colonial breeding [31]. During breeding, they must forage for both self-maintenance and offspring provisioning [32]. Therefore, it is assumed that seabirds should target prey that is of a high-energy content in order to reduce foraging effort. However, little is known of the strategies individuals adopt when high-calorific prey is not available [10,33]. In addition, many breeding seabirds, particularly penguins (family: Spheniscidae), display a narrow foraging range due to regular nest attendance patterns and, thus, forage under both spatial and temporal constraints [34]. Understanding how individuals find and consume prey during these important periods is essential to predicting how they may respond to alterations in the composition of prey communities, which ultimately impacts their reproductive success and population survival.

The little penguin (*Eudyptula minor*) is an important consumer in the marine ecosystems of southeastern Australia, a highly variable region with some of the most rapidly warming waters in the world [35]. Due to its high cost of transport and small size, the little penguin has one of the most reduced foraging ranges of any penguin [36]. While some have suggested negative consequences in breeding success due to changes in pelagic prey availability [37], few studies have investigated how the immediate prey-field influences little penguin foraging decisions [38]. Understanding how individuals maximize their foraging success when faced with prey varying in quantity and calorific quality may provide insights into how anticipated changes to prey availability in this region may impact predator–prey interactions [39].

The aims of this study were to determine in little penguins the relationship between foraging effort and prey-field factors: (i) prey type and (ii) patch density. According to foraging theory [40,41], it is hypothesized that individuals will forage longer at prey patches of higher calorific density. It is predicted that animals should be able to distinguish between prey types to choose those with the greatest profitability. Therefore, penguins should adjust their foraging effort according to the calorific value of their prey [42–44].

## 2. Methods

To increase the likelihood of detecting varying types and densities of prey, the study was conducted at two breeding colonies in southeastern Australia: London Bridge (LB, 38°62′ S, 142°93′ E) and Gabo Island (GI, 37°56′ S, 149°91′ E). During the 2014–2015 (October–January) breeding period, adult penguins

rearing small chicks were captured in their nest burrows and weighed in a cloth bag using a spring balance (±10 g).

Individuals were instrumented with a video data logger (Catnip Technologies Ltd, USA, $30 \times 40 \times 15$ mm, 20 g, $400 \times 400$ pixels at 30 frames $s^{-1}$) programmed to record continuously on a duty cycle of 15 min every hour, and a GPS (Mobile Action Technology, I-gotU, GT-120, $44.5 \times 28.5 \times 13$ mm, 17 g) programmed to sample location every 2 min. The devices were attached along the dorsal midline using Tesa® tape (4651), with the camera facing forward. Together, they weighed 3% of the average little penguin body mass (1109 ± 17 g) and were less than 1% cross-sectional surface area. Individuals were recaptured in their nest burrows and the devices were removed after a single foraging trip. The morphometrics bill depth, bill length, head length and flipper length were measured using a Vernier calliper and ruler (±0.1 mm and 1 mm, respectively). Bill length was used to determine sex with males generally presenting with a larger bill depth than females [45]. Handling periods at deployment and retrieval did not exceed 10 min and measurements were completed on retrieval to minimize handling at deployment. After handling, individuals were released back into their nests to resume normal activity.

## 2.1. Data processing and statistical analyses

Unless otherwise stated, all data analyses were conducted in the R Statistical Environment (v. 4.0.2) [46]. The GPS location data were filtered to remove erroneous fixes that exceeded the maximum average horizontal travel speed of 7.2 m s$^{-1}$ [36] and the foraging trip metrics range (km) and total duration (h) were determined using the *trip* package [47].

Video data were viewed and analysed through a custom annotation sheet created in Solomon Coder (Budapest, Hungary, v. 16.06.26). Video data was used to determine prey type, prey capture events and the beginning and end of a foraging bout. For each foraging bout, prey were identified in the video data to the lowest taxonomic level possible with aid of a fish identification guide [48]. It was not always possible to determine total patch abundance as the whole prey patch was not always visible. Therefore, an index of prey patch density was calculated for each fish prey patch from a random sample of 10 still images from each foraging bout. A rectangle based on three body lengths × two body lengths of the observed fish was overlaid on the still images and all fish within that rectangle were counted. Prey items on the fringe of the rectangle were included in counts if more than 50% of their body was inside the area. This was repeated three times for each image with rectangles being placed at random on the bait ball (figure 1). The physical height and length of each polygon was then estimated from the body length of each fish species from literature [48]. Larval fish were present in video footage and, as it was not possible to determine the species, the average larval fish size for all Clupeiformes species present in this study was used [38]. These values were used to determine the numerical window size and calculate estimates of density (number of prey m$^{-2}$) which was then standardized to control for different window sizes between prey types using the following equation:

$$D_{scaled} = \frac{x - \overline{PT}}{\sigma_{PT}},$$

where $D_{scaled}$ is the scaled density value, $x$ is the non-scaled value, the mean ($\overline{PT}$) and the standard deviation ($\sigma$) for prey type density (*PT*).

As an indication of prey quality, energetic content of prey types was estimated from a previous study [38]. These gross energy (kJ) estimates were multiplied by the densities and scaled using the same above formula. These were used to provide ecological context for foraging decisions. Penguins are known to forage on prey patches through series of dives, called a dive bout [49]. Directly after a bout of foraging, individuals generally display either resting or travelling periods. Hence, a foraging bout was considered to have ceased if an individual spent greater than 1 min resting or preening on the surface or if there were consecutive shallow travelling dives and no prey were encountered. Patch residency time (PRT, min) was used as a proxy for effort and was defined as the beginning of the first dive in which the individual encountered a prey patch to the end of the last dive in a patch. Foraging bouts were removed from further analysis if it was not possible to identify the beginning and end of the event (e.g. if the video period ended before the individual ceased foraging).

Linear mixed effects (LME) models were used to determine the factors influencing PRT using the *lme4* package [50]. Initial comparisons revealed no differences in PRT between sexes or colonies and so data were pooled. The assumptions of homoscedasticity and normal distribution of residuals were tested using Levene's tests and visual inspection of qq-plots, respectively. This revealed PRT was positively

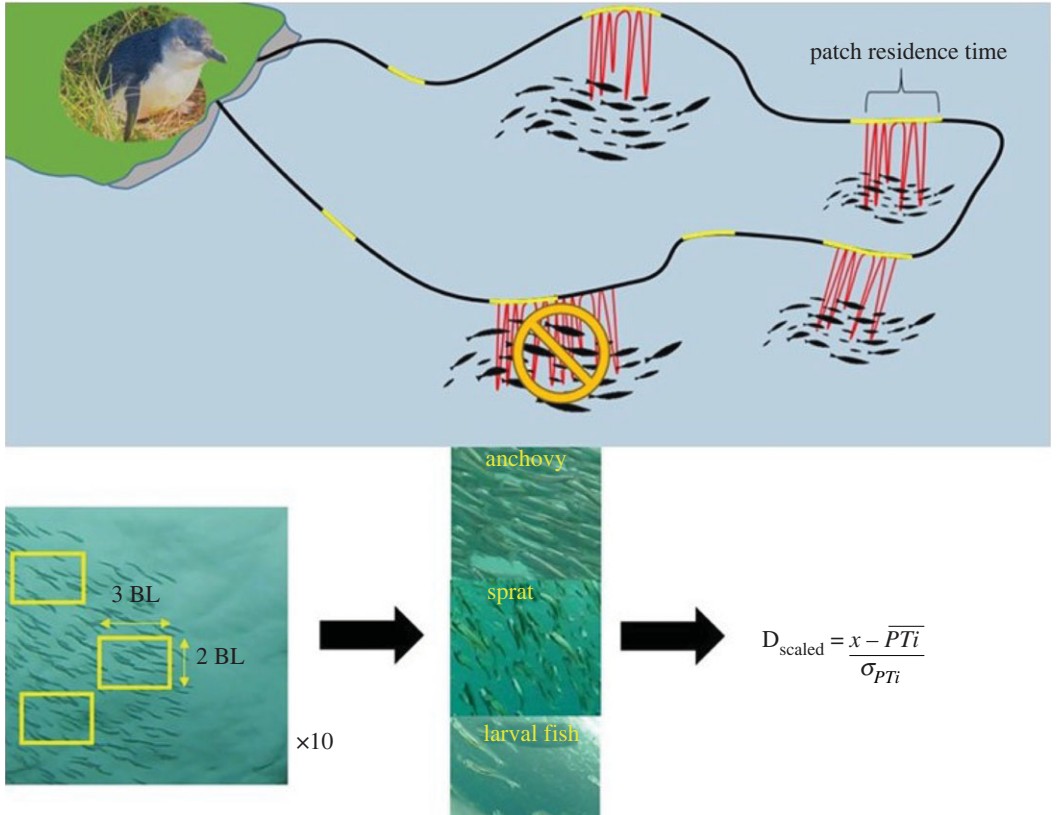

**Figure 1.** Schematic for determining prey abundance and patch residence time from animal-borne video camera data obtained from free-ranging little penguins. Little penguins were tracked with GPS (black line) and depth recorder (red lines indicating dives). Video data (yellow lines) were collected over a foraging trip at increments of 15 min every hour. Patch residence time, a proxy for foraging effort, was calculated from the beginning to the end of the first and last dive at a patch. Only prey patches that commenced and concluded during a video recording period were used. Number of fish in a window size of $3 \times 2$ body lengths (BL) was used as an index of abundance for each fish prey type and converted to centimetres based on average BL values. These values were subtracted from the mean density and divided by the standard deviation to obtain a scaled numerical index of patch abundance ($D_{scaled}$).

skewed, which was subsequently improved through log transformation. The model included prey type, $D_{scaled}$ and an interaction between the two variables. To determine if effort is influenced by the energetic quality of a patch, a second LME was performed which included prey type, the calorific density and an interaction between the two variables. All models included individual as a random factor to account for repeated measures. Model selection was undertaken using the *MuMIn* package [51] to determine the most parsimonious combination of predictor variables. The best model was determined as the one with the lowest Akaike information criterion, corrected for small sample sizes (AICc) and candidate models with a ΔAICc of less than 4 are presented as the likely models for explaining PRT [52]. Unless otherwise stated, data are presented as mean ± s.e.

## 3. Results

A total of 22 little penguins (11 from each colony) were instrumented for a single foraging trip, which lasted on average 15 ± 0.4 h during which individuals travelled a total of 45.06 ± 3.8 km near the colony (range: 18.5 ± 1.4 km; table 1). Ten of the 22 individuals gained between 10 and 130 g after a single foraging trip. However, as birds were captured after they had returned to their burrows, these values are an imprecise representation of foraging success as it is possible that they had already fed their chicks.

Video data provided on average 4.7 ± 1.1 h of the foraging trip, accounting for approximately 25% of each individual trip. A total of 320 prey patches were identified, where penguins completed 776 dives (range: 1–36 dives per patch) and consumed a total of 620 prey items. Four main prey species were identified from the video data: southern anchovy, *Engraulis australis*; sandy sprat, *Hyperlophus vittatus*; juvenile Clupeiformes (hereafter referred to as larval fish) and jellyfish, *Cyanea* spp. (figure 1).

**Table 1.** Summary of camera and GPS deployments on little penguins from two colonies (Gabo Island, GI and London Bridge, LB) in southeastern Australia. The number of video intervals and associated prey patches, total captures, dives within patches and the average patch capture rate are provided.

| colony | ID | sex | deployment date | deployment mass (kg) | retrieval mass (kg) | video intervals (N) | number of patches | total captures | total dives | average capture rate |
|---|---|---|---|---|---|---|---|---|---|---|
| GI | LP01 | F | 13/01/2015 | 1.1 | 1.08 | 18 | 70 | 115 | 86 | 1.6 |
| GI | LP02 | F | 13/01/2015 | 1.01 | 1.04 | 12 | 6 | 9 | 14 | 1.5 |
| GI | LP03 | M | 29/11/2014 | 1.16 | 1.23 | 17 | 9 | 23 | 30 | 2.6 |
| GI | LP04 | M | 29/11/2014 | 1.24 | 1.21 | 13 | 3 | 2 | 3 | 0.67 |
| GI | LP05 | F | 30/11/2014 | 1.11 | 1.19 | 18 | 32 | 26 | 32 | 0.81 |
| GI | LP06 | F | 30/11/2014 | 1.12 | 1.25 | 14 | 14 | 8 | 14 | 0.57 |
| GI | LP07 | M | 30/11/2014 | 1.14 | 1.24 | 12 | 5 | 11 | 18 | 2.2 |
| GI | LP08 | M | 13/01/2015 | 1.12 | 1.09 | 13 | 31 | 77 | 59 | 2.5 |
| GI | LP09 | M | 13/01/2015 | 1.03 | 1 | 12 | 35 | 82 | 62 | 2.3 |
| GI | LP10 | M | 13/01/2015 | 1.08 | 1.05 | 11 | 15 | 18 | 16 | 1.2 |
| GI | LP11 | F | 13/01/2015 | 1.09 | 1.06 | 12 | 11 | 40 | 63 | 3.6 |
| LB | LP12 | M | 10/12/2014 | 1.07 | 1.16 | 11 | 29 | 31 | 35 | 1.1 |
| LB | LP13 | F | 09/11/2014 | 1.01 | 1.09 | 15 | 1 | 4 | 20 | 4 |
| LB | LP14 | F | 10/12/2014 | 1 | 1.02 | 18 | 7 | 21 | 28 | 3 |
| LB | LP15 | M | 28/12/2014 | 1.17 | 1.11 | 19 | 5 | 30 | 31 | 6 |
| LB | LP16 | M | 27/12/2014 | 1.14 | 1.15 | 15 | 4 | 5 | 4 | 1.3 |
| LB | LP17 | M | 27/12/2014 | 1.2 | 1.14 | 12 | 8 | 8 | 8 | 1 |
| LB | LP18 | F | 27/12/2014 | 1.07 | 1.05 | 7 | 14 | 15 | 14 | 1.1 |
| LB | LP19 | F | 27/12/2014 | 0.94 | 0.99 | 20 | 6 | 6 | 6 | 1 |
| LB | LP20 | F | 28/12/2014 | 1.03 | 1.07 | 17 | 5 | 14 | 43 | 2.8 |
| LB | LP21 | M | 27/12/2014 | 1.14 | 1.17 | 16 | 8 | 57 | 126 | 7.1 |
| LB | LP22 | M | 28/12/2014 | 1.06 | 1.02 | 19 | 2 | 18 | 64 | 9 |

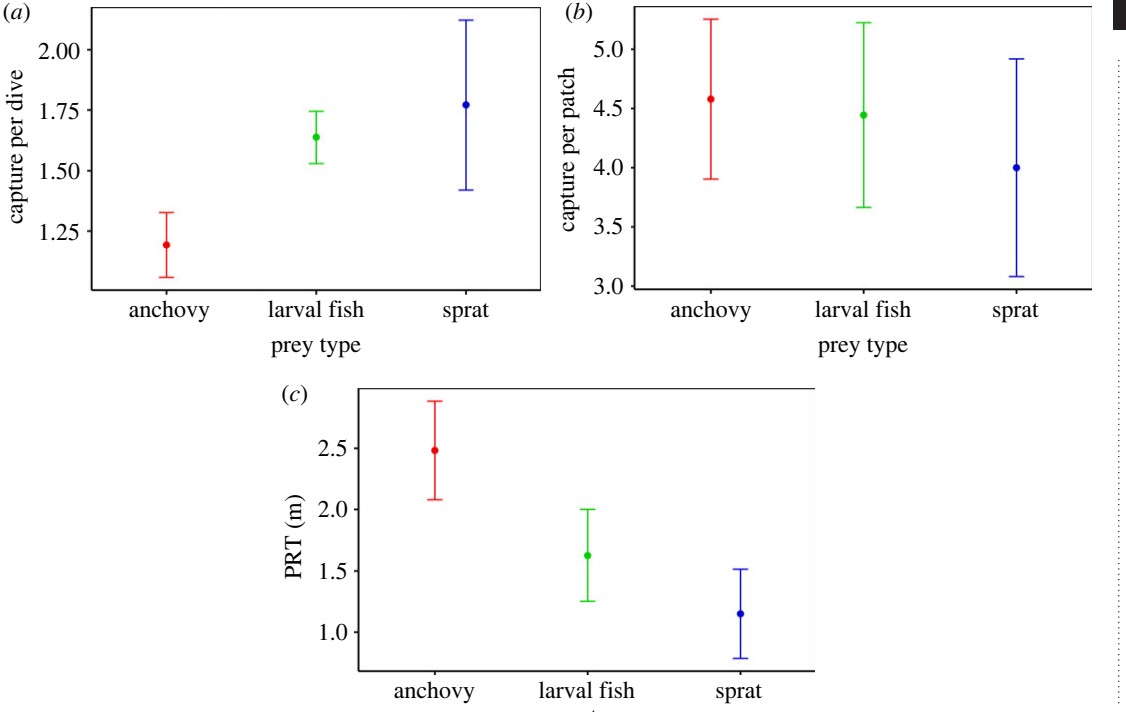

**Figure 2.** Average captures per dive (*a*), per prey patch (*b*) and PRT (*c*) for prey types (anchovy, larval fish and sprat) encountered by little penguins.

**Table 2.** Summary of prey events by prey type, the proportion of prey captures which were categorized as occurring as part of a foraging bout, range of observed prey patch densities, capture rates and estimated energy content.

| prey type | events (N) | total prey captures | patch events (%) | density (m²) | captures per dive ± s.e. | captures per patch ± s.e. | energy content (kJ g⁻¹) |
|---|---|---|---|---|---|---|---|
| anchovy | 93 | 241 | 40.4 | 0.04–38.4 | 1.2 ± 0.1 | 4.5 ± 0.7 | 5.2 |
| sprat | 29 | 59 | 34.5 | 2.6–25.2 | 1.8 ± 0.3 | 4 ± 0.9 | 5 |
| larval fish | 95 | 218 | 37.9 | 5.5–29.2 | 1.6 ± 0.1 | 4.4 ± 0.8 | 2.2 |

Numerical and calorific densities were the highest during encounters of anchovy, followed by sprat and larval fish (table 2). The lowest numerical and calorific densities were of jellyfish as they do not school closely together. These were removed from further analyses ($N = 103$) along with prey captures of solitary prey items (representing 29% of all prey captures). Individuals consumed on average $4.7 \pm 1$ fish per prey patch and spent between 0.5 and 9.6 min foraging at a single patch (mean $12.1 \pm 0.2$ min). These metrics seemed to be influenced by prey type as penguins spent longest at patches of anchovy, followed by larval fish and sprat (figure 2). Although prey captures per dive were highest for larval fish and sprat, there was no difference between average number of prey captured at the patch scale (figure 2). There were no instances where the prey patch was consumed entirely, with individuals ceasing foraging when the prey patch had dispersed.

The most parsimonious model explaining PRT included numerical patch density, prey type and an interaction between the two variables (table 3). This model accounted for more than 81% of the variation observed in PRT and was the only candidate model with a ΔAICc of less than 4. In general, penguins spent significantly longer at prey patches of higher numerical densities of prey (figure 3). However, the interaction term indicated that the relationship between numerical density and PRT was prey specific with individuals spending longer at anchovy patches than sprat and larval fish (table 3). When accounting for the calorific value in density estimates, the model followed a similar overall

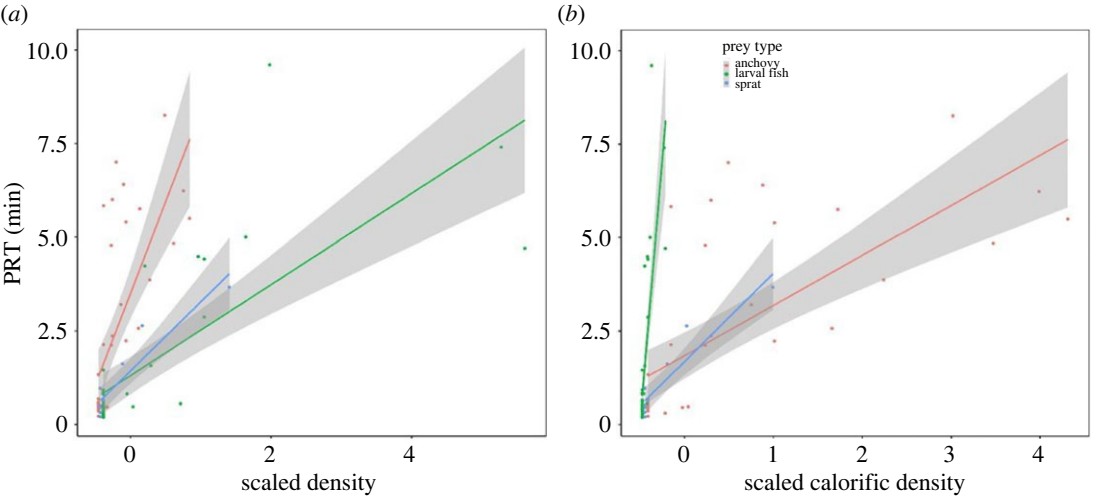

**Figure 3.** PRT (min) for scaled numerical density of each prey type (*a*) and scaled calorific density (*b*). Shaded area represents the 95% confidence interval and black line indicates overall relationship between PRT and $D_{scaled}$.

**Table 3.** Model summary table for foraging effort (PRT) in little penguins. The predictors: numerical density ($D_{scaled}$), PT (prey type) and calorific density (kJ $D^{-1}$) were all retained in the most parsimonious models. Significant predictor effects ($p < 0.05$) are indicated in italics.

| response | predictors | estimate | s.e. | *t*-value | CI | *p* |
|---|---|---|---|---|---|---|
| PRT | intercept | 0.98 | 0.19 | 5.16 | 0.62, 1.34 | *<0.001* |
| | $D_{scaled}$ | 2.07 | 0.39 | 5.26 | 1.32, 2.83 | *<0.001* |
| | PT(larval fish) | −0.84 | 0.22 | −3.76 | −1.28, −0.41 | *<0.001* |
| | PT(sprat) | −0.69 | 0.34 | −2.06 | −1.35, −0.04 | *0.04* |
| | $D_{scaled}$ * PT(larval fish) | −1.61 | 0.40 | −4.07 | −2.37, −0.86 | *<0.001* |
| | $D_{scaled}$ * PT(sprat) | −1.05 | 0.54 | −1.93 | −2.09, 0.01 | 0.06 |
| PRT | intercept | 0.27 | 0.19 | 1.40 | −0.10, 0.64 | 0.1 |
| | kJ $D^{-1}$ | 0.57 | 0.11 | 5.26 | 0.36, 0.77 | *<0.001* |
| | PT(larval fish) | 4.72 | 1.12 | 4.20 | 2.57, 6.86 | *<0.001* |
| | PT(sprat) | 0.16 | 0.34 | 0.46 | −0.52, 0.81 | 0.6 |
| | kJ $D^{-1}$ * PT(larval fish) | 10.10 | 2.46 | 4.1 | 5.41, 14.80 | *<0.001* |
| | kJ $D^{-1}$ * PT(sprat) | 0.74 | 0.49 | 1.50 | −0.20, 1.71 | 0.1 |

trend across all prey types with individuals spending longer at patches of higher calorific densities. However, the interaction term revealed that penguins spent significantly longer at densities of larval fish even though it was of lower calorific density/quality than the other prey types (figure 3).

## 4. Discussion

Being visual predators, little penguins are constrained to forage only during daylight hours, with all individuals in the present study returning to the colony at sunset [53]. Consistent with previous studies [36,54], individuals in the present study displayed narrow foraging ranges during the breeding season, highlighting their spatial and temporal foraging constraints. Subject to these constraints, penguins maximized their capture rate over short periods by seeking to exploit patches where prey were present in high densities, irrespective of the potential calorific value.

Optimal foraging theory suggests that individuals should make decisions to forage in a way that uses the least amount of energy for the most amount of energy gained [30]. When prey is distributed throughout the environment in discrete patches, such as in the present study, individuals are expected

to leave a patch when the rate of gain drops to that of the average gain within the environment [40,41]. As prey types can vary in calorific quality, effort at a single patch should change according to prey type [55]. However, individuals in the present study did not adjust their foraging decisions as expected. This may be due to the spatio-temporal constraints imposed on them during breeding.

Outside the breeding season, little penguins have been recorded spending between 2 and 49 days at sea and foraging at maximum distances of 147 km from the colony [56]. In comparison, during early chick-rearing, foraging range and duration are highly restricted, with individuals typically alternating between guarding the chick in the nest and foraging each day. In the present study, the spatio-temporal restrictions on the foraging range of penguins may have resulted in the unbiased behaviour towards all prey types. Particularly since, during early breeding, individuals were unable to travel further afield for better quality prey and still meet offspring provisioning requirements.

Although little penguins are considered to be generalist foragers [56], it was predicted that individuals should still differentiate between high- and low-quality prey types in order to maintain the necessary time-energy balance. However, in the present study, patch numerical density was an important factor in predicting effort. Increased residency time at patches of higher density may also imply that penguins do not have prior knowledge of the habitat quality. Indeed, little penguins display low foraging site fidelity, rarely overlapping in foraging locations on consecutive trips [57]. This probably reflects the highly dynamic nature of the marine environment and the mobility of pelagic prey.

When accounting for gross energy values in density estimates, individuals spent longer at patches of larval fish than anchovy or sprat. Although larval fish are of lower calorific quality, it is possible penguins may spend a greater time at these patches because fish at this life stage are less mobile in comparison with adult fish and, thus, are easier to capture [58]. Indeed, individuals at larval fish patches had a significantly higher capture rate per dive than anchovy, probably reflecting their faster handling/processing times. As larval fish had the greatest numerical densities in the present study, this may have enabled penguins to consume larger quantities of these prey before the patches dispersed. The significantly higher capture rate per dive observed when feeding on larval fish could indicate that penguins compensate for the lower calorific value of these prey items by consuming larger quantities per dive. This could ultimately result in a similar calorific gain across all prey types.

The increased time at patches of larval fish may reflect a trade-off between capture effort and energetic gain. Exerting less effort in capturing prey may result in overall higher energy gained and could also explain why penguins were observed to consume gelata [59]. Minimizing effort is important for seabirds such as penguins, where the cost of transport is higher than for flying seabirds [60,61]. In addition, individuals may stay longer at prey patches, especially if time between foraging bouts is unpredictable [62]. Such responses have also been shown in terrestrial species experiencing constraints in their foraging time [63]. In the present study, PRT was used as a metric of foraging effort which considers time, not energy use. Although time has been shown to correlate well with energy use [60], future studies should consider using accelerometers to better understand energy expenditure and gain associated with foraging on certain prey types.

The present study revealed the fine-scale foraging decisions made by little penguins in relation to their immediate prey-field. The results indicated that individuals are influenced by the density of prey at a location rather than its potential calorific value. This probably reflects the spatio-temporal constraints associated with breeding and the dynamic nature of pelagic prey in the marine environment. Individuals may choose prey patch quantity over calorific quality during their time- and distance-limited foraging trips. However, it is important to note that the foraging decisions observed in the present study may not be representative of the species' annual cycle. Indeed, several seabird species have been found to extend their range and use alternative strategies outside of the breeding season [64]. Future studies should consider investigating how the prey-field influences foraging behaviour outside of the breeding season.

Ethics. The animal use procedures used in the present study were approved by the Deakin University Animal Ethics Committee (B25-2016 01/08/2016) and conducted under Wildlife Research Permit (10008044) of the Department of Environment, Land, Water and Planning (Victoria).

Data accessibility. The data that support the findings of this study are openly available in Zenodo at https://doi.org/10.5281/zenodo.6403771 [65] and https://doi.org/10.5281/zenodo.6413352 [66]. R code and the information used to generate modelling results are available as electronic supplementary material [67].

Authors' contributions. G.J.S.: conceptualization, data curation, formal analysis, investigation, methodology, project administration, visualization, writing—original draft and writing—review and editing; J.P.Y.A.: conceptualization, methodology, project administration, resources, supervision, validation, writing—original draft and writing—review and editing.

All authors gave final approval for publication and agreed to be held accountable for the work performed therein.
Conflict of interest declaration. We declare we have no conflict of interest declaration.
Funding. This study was supported by Deakin University.
Acknowledgements. We thank Parks Victoria for providing logistical support throughout this project and the many volunteers who assisted in data collection.

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
