## [Peer Review File · Royal Society Open Science]

Review History

RSOS-211171.R0 (Original submission)

Review form: Reviewer 1

Is the manuscript scientifically sound in its present form?

Yes

Are the interpretations and conclusions justified by the results?

Yes

Is the language acceptable?

Yes

Do you have any ethical concerns with this paper?

No

Have you any concerns about statistical analyses in this paper?

No

Recommendation?

Major revision is needed (please make suggestions in comments)

Comments to the Author(s)

The paper “Quantity over quality? Prey-field characteristics influence the foraging decisions of little penguins (*Eudyptula minor*)” strives to advance the use of animal-borne video to inform important questions on predator-prey interactions in the marine environment. The paper reports positive relationships between time spent in prey patches and a simple index of areal prey density derived from 2D video images. The authors infer that little penguin foraging effort, which is strongly constrained by the peculiarities of central place foraging, best relates to apparent prey density and not prey quality as measured by energetic content. I like and appreciate this novel approach to investigating predator-prey interactions and look forward to seeing this published. However, there are gaps in the methods and a general lack of clarity on assumptions made to facilitate the analysis. Fixing these weaknesses is needed to make the research more repeatable and to strengthen confidence in the main results.

The methods should better describe how the foraging behaviors of the birds were quantified. For example, a brief description of how the videos were analyzed to quantify predation and patch entry/exit is needed. Specifically, were multiple annotations made (and how) so that you can present an error on your estimate of the number of prey captured? What criteria were used to select images for counting prey items? How was the determination of patch entry/exit quantified? Were these periods of foraging that define PRT bounded by clear period of transit or failed foraging dives, or some other criteria? You note that patch dispersal often signaled the end of time in the patch. What does patch dispersal look like with respect to the apparent density estimates provided during foraging? What assumptions were needed to make this determination of finite foraging time in the patch given that only 15 minutes of foraging efforts were observed? Another consideration for the methods is how prey sizes and prey depth distributions affect time spent in a particular patch of prey. On the former, as gape-limited predators dependent on the capture of single prey items, one at a time, increased foraging time in a patch could depend on the time required to fill the stomach with large versus small items. It would be helpful to document any relationship in the data between the density index and prey sizes encountered and include an index of prey size (perhaps determined from the image, similar to the inclusion of apparent density from the image) in the models that seek to explain foraging duration in the patch. On the latter, I realize you have no data on depth, but perhaps you can speculate on whether particular prey are found at different depths and whether different depth distributions affects time available to little penguins for foraging (not to mention the definition of PRT you have adopted which includes all behaviors during a bout of diving). Additionally, it may be worth discussing how differences in anti-predator behaviors/morphologies of the different prey items could affect PRT.

Specific Comments

Introduction (and throughout): with respect to the data on time spent in prey patches, the term “foraging success” should be changed to “foraging effort”, since the two are not necessarily equivalent but seem to be used interchangeably here.

Methods

Were all the videos recorded simultaneously, so that all individuals can be assumed to have reasonably encountered the same prey aggregations? Providing some indication of the temporal separation of sampling by individuals will be helpful for considering how changes in prey availability over a season may affect foraging behaviors.

Line 86: clarify what was measured with calipers, please

Line 107: do you know the species of larval fish in the imagery? The text here mentioning the dimensions of the polygons suggests you do. If so, replace “larval fish” with the appropriate species names. Else, please describe how a length for generic larval fishes was estimated.

Line 104-107: How did you translate fish lengths from the literature to appropriately sized polygons on the frame? This needs clarification. Also, did you count everything within those

small polygons, or only clearly recognizable prey species? Did you count only individuals that were completely within the frame, or also those crossing the frame to some degree? Much more clarity on methods and assumptions is required to understand how you estimated areal density here.

Line 110-113 – presumably these mean and standard deviation statistics refer to the summaries of the raw density estimates from the 3 replicate counts of fish in the frame on each of 10 images per dive on each fish type (N=?) so that you represent the scaled density as a variable with zero mean and unit variance? If so, are the axes in Figure 2 correct? Please clarify. Also, please clearly define the acronyms used.

Line 116-117: how was the determination of patch entry/exit quantified? Were these bounded by clear period of transit or failed foraging dives? Also, please present units for the PRT metric here and throughout.

Lines 117-119- this assumption may be appropriate if prey depth distributions (and time spent on ascent/descent) is similar across prey patches. If not, is the PRT metric contaminated by variation in behaviors not directly related to foraging? Why not quantify PRT on a per-dive basis that includes only the time within the actual prey patch?

Results

It will help to understand the efficiency of the methods if you can provide additional, general summaries of the data used in the study. For example, from the 22 deployments, how many 15min video segments were used and what percent of the total recorded time is included in PRT? Of the video used, did all the individuals exhibit foraging in different prey types or were some specialized on specific prey only? Consider adding some indication of predator foraging success (e.g. mean fish eaten per dive could be added to Table 1, rather than the S1 figure)

Discussion

Line 170-172: The authors state the birds showed a reduction in distance and duration. Of what?

Line 184-186- or because it takes longer to fill up on small prey items?

Figure 2 – can you show the scaled density data in the plot? It is needed to help evaluate the model fit.

Decision letter (RSOS-211171.R0)

Dear Miss Sutton

The Editors assigned to your paper RSOS-211171 "Quantity over quality? Prey-field characteristics influence the foraging decisions of little penguins (*Eudyptula minor*)." have now received comments from reviewers and would like you to revise the paper in accordance with the reviewer comments and any comments from the Editors. Please note this decision does not guarantee eventual acceptance.

We do not generally allow multiple rounds of revision so we urge you to make every effort to fully address all of the comments at this stage. If deemed necessary by the Editors, your

manuscript will be sent back to one or more of the original reviewers for assessment. If the original reviewers are not available, we may invite new reviewers.

Please submit your revised manuscript and required files (see below) no later than 21 days from today's (ie 15-Nov-2021) date. Note: the ScholarOne system will 'lock' if submission of the revision is attempted 21 or more days after the deadline. If you do not think you will be able to meet this deadline please contact the editorial office immediately.

on behalf of Dr Agustina Gómez-Laich (Associate Editor) and Pete Smith (Subject Editor)
openscience@royalsociety.org

Associate Editor Comments to Author (Dr Agustina Gómez-Laich):

Associate Editor: 1

Comments to the Author:

In this study, authors take advantage of the information provided by animal-borne video cameras to examine if Little penguins (*Eudyptula minor*) foraging behaviour is influenced by prey type and patch density. The paper is well written and presents valuable information which adds to the body of literature on how prey characteristics influence predators' decisions in their natural environment however authors should address several aspects before the manuscript could be published. Several sections of the methodology need to be better explained to understand how results were obtained. Additionally, it would greatly improve the manuscript if in the introduction authors present both the main hypothesis and the associated predictions. Finally, I suggest discussing more the fact that the obtained results do not consider the energy expenditure associated to capturing different prey types. High caloric prey may not be so profitable if the energy expenditure associated to capture it is high. For example, it will greatly contribute to the Ms if authors present an estimation of how costly it is for a little penguin to capture larval fish, anchovy, and sprat.

Specific comments.

Introduction

Line 44. In addition to GPS and time-depth recorders, there are other methods such as the imasen, stomach temperature sensors and accelerometry that allow estimating prey capture events. I suggest incorporating this information.

Line 52. Please incorporate more references here. For example:

- 1) Akiyama Y, Akamatsu T, Rasmussen MH, Iversen MR, Iwata T, Goto Y, et al. (2019) Leave or stay? Video-logger revealed foraging efficiency of humpback whales under temporal change in prey density. *PLoS ONE* 14(2): e0211138. <https://doi.org/10.1371/journal.pone.0211138>
- 2) Watanabe YY, Ito M, Takahashi A. Testing optimal foraging theory in a penguin-krill system. *Proc Biol Sci.* 2014 Jan 29;281(1779):20132376. doi: 10.1098/rspb.2013.2376. PMID: 24478293; PMCID: PMC3924065.

Line 79. It would greatly improve the manuscript if not only the hypothesis but also the derived predictions are presented.

Methods.

Line 90. Did cameras record continuously for 15 minutes every hour? I believe they did but please make this clear.

Line 93. Please incorporate penguins' body mass. How much time did the instrumentation procedure last? Were all penguins instrumented in the same period? You said instrumentations were performed between October and January. It would be nice to incorporate a table with the instrumentation date of each animal. How were animals captured? Did you have a control group in order to determine if the presence of a camera and GPS had an effect on penguins' foraging behaviour (i.e. foraging trip duration)? After instrumentation, did birds continue breeding normally? Please incorporate these pieces of information.

Line 96. You state that all analyses were performed with R. Even video visualization?

Line 101. It is not clear why 10 still images from each dive were selected. Couldn't it be better to relate the number of still images per dive to the duration of the dive? Or were all recorded dives similar in duration?

Line 106. Were these rectangles placed at random on the image?

Line 107. You say that the physical height and length of each polygon was estimated from the body length of each fish species from the literature. Which body length was considered? The mean adult body length of each species? And if the fish belonged to a different stage (larval for example)?

Line 117. How did you define when a penguin was entering and leaving a patch? Please clarify this aspect.

Line 119. How did you do when the bird entered a patch but the video recording stopped before it left? This is briefly explained in figure 1. Please incorporate this information into the methods section.

Line 120. PRT instead of (PRT)

Stats. Which library did you employ to perform the models, and which library did you use for model selection? Please incorporate this information.

It would be interesting to incorporate some depth data in order to see if decisions are influenced by this variable. Did you have depth data? Could you estimate it by using the time it took little penguins to reach the bottom phase of a dive?

Results.

Line 147-154. In relationship to the previous comments, do penguins capture the different prey types at similar depths? Couldn't it be that anchovy and sprat shoals are more easily broken than larval fish shoals so penguins stay in the larval fish shoals for longer? Authors discuss this briefly, I suggest discussing more in detail how differences in prey behaviour may affect predators decisions.

Discussion.

I suggest discussing more the results in the context that what authors report does not consider the energy expenditure associated with capturing different prey types. High caloric prey may not be so profitable if the energy expenditure associated to capture it is high. Not having estimations of the energy expenditure associated to capturing different prey types is a key limitation of the study and something that could be measured in the future for example by means of accelerometry.

Line 170-172. A reduction in distance and time in relation to the non-foraging period? Is this what you are trying to say? Please rephrase the sentence.

Line 202. Eliminate "his"

Reviewer comments to Author:

Reviewer: 1

Comments to the Author(s)

The paper "Quantity over quality? Prey-field characteristics influence the foraging decisions of little penguins (*Eudyptula minor*)" strives to advance the use of animal-borne video to inform important questions on predator-prey interactions in the marine environment. The paper reports positive relationships between time spent in prey patches and a simple index of areal prey density derived from 2D video images. The authors infer that little penguin foraging effort, which is strongly constrained by the peculiarities of central place foraging, best relates to apparent prey density and not prey quality as measured by energetic content. I like and appreciate this novel approach to investigating predator-prey interactions and look forward to seeing this published. However, there are gaps in the methods and a general lack of clarity on assumptions made to facilitate the analysis. Fixing these weaknesses is needed to make the research more repeatable and to strengthen confidence in the main results.

The methods should better describe how the foraging behaviors of the birds were quantified. For example, a brief description of how the videos were analyzed to quantify predation and patch entry/exit is needed. Specifically, were multiple annotations made (and how) so that you can present an error on your estimate of the number of prey captured? What criteria were used to select images for counting prey items? How was the determination of patch entry/exit quantified? Were these periods of foraging that define PRT bounded by clear period of transit or failed foraging dives, or some other criteria? You note that patch dispersal often signaled the end of time in the patch. What does patch dispersal look like with respect to the apparent density estimates provided during foraging? What assumptions were needed to make this determination of finite foraging time in the patch given that only 15 minutes of foraging efforts were observed? Another consideration for the methods is how prey sizes and prey depth distributions affect time spent in a particular patch of prey. On the former, as gape-limited predators dependent on the capture of single prey items, one at a time, increased foraging time in a patch could depend on the time required to fill the stomach with large versus small items. It would be helpful to document any relationship in the data between the density index and prey sizes encountered and include an index of prey size (perhaps determined from the image, similar to the inclusion of

apparent density from the image) in the models that seek to explain foraging duration in the patch. On the latter, I realize you have no data on depth, but perhaps you can speculate on whether particular prey are found at different depths and whether different depth distributions affects time available to little penguins for foraging (not to mention the definition of PRT you have adopted which includes all behaviors during a bout of diving). Additionally, it may be worth discussing how differences in anti-predator behaviors/morphologies of the different prey items could affect PRT.

Specific Comments

Introduction (and throughout): with respect to the data on time spent in prey patches, the term “foraging success” should be changed to “foraging effort”, since the two are not necessarily equivalent but seem to be used interchangeably here.

Methods

Were all the videos recorded simultaneously, so that all individuals can be assumed to have reasonably encountered the same prey aggregations? Providing some indication of the temporal separation of sampling by individuals will be helpful for considering how changes in prey availability over a season may affect foraging behaviors.

Line 86: clarify what was measured with calipers, please

Line 107: do you know the species of larval fish in the imagery? The text here mentioning the dimensions of the polygons suggests you do. If so, replace “larval fish” with the appropriate species names. Else, please describe how a length for generic larval fishes was estimated.

Line 104-107: How did you translate fish lengths from the literature to appropriately sized polygons on the frame? This needs clarification. Also, did you count everything within those small polygons, or only clearly recognizable prey species? Did you count only individuals that were completely within the frame, or also those crossing the frame to some degree? Much more clarity on methods and assumptions is required to understand how you estimated areal density here.

Line 110-113 – presumably these mean and standard deviation statistics refer to the summaries of the raw density estimates from the 3 replicate counts of fish in the frame on each of 10 images per dive on each fish type (N=?) so that you represent the scaled density as a variable with zero mean and unit variance? If so, are the axes in Figure 2 correct? Please clarify. Also, please clearly define the acronyms used.

Line 116-117: how was the determination of patch entry/exit quantified? Were these bounded by clear period of transit or failed foraging dives? Also, please present units for the PRT metric here and throughout.

Lines 117-119- this assumption may be appropriate if prey depth distributions (and time spent on ascent/descent) is similar across prey patches. If not, is the PRT metric contaminated by variation in behaviors not directly related to foraging? Why not quantify PRT on a per-dive basis that includes only the time within the actual prey patch?

Results

It will help to understand the efficiency of the methods if you can provide additional, general summaries of the data used in the study. For example, from the 22 deployments, how many 15min video segments were used and what percent of the total recorded time is included in PRT? Of the video used, did all the individuals exhibit foraging in different prey types or were some specialized on specific prey only? Consider adding some indication of predator foraging success (e.g. mean fish eaten per dive could be added to Table 1, rather than the S1 figure)

Discussion

Line 170-172: The authors state the birds showed a reduction in distance and duration. Of what?

Line 184-186- or because it takes longer to fill up on small prey items?

Figure 2 – can you show the scaled density data in the plot? It is needed to help evaluate the model fit.

===PREPARING YOUR MANUSCRIPT===

If you have been asked to revise the written English in your submission as a condition of publication, you must do so, and you are expected to provide evidence that you have received language editing support. The journal would prefer that you use a professional language editing service and provide a certificate of editing, but a signed letter from a colleague who is a fluent speaker of English is acceptable. Note the journal has arranged a number of discounts for authors using professional language editing services (<https://royalsociety.org/journals/authors/benefits/language-editing/>).

===PREPARING YOUR REVISION IN SCHOLARONE===

- An individual file of each figure (EPS or print-quality PDF preferred [either format should be produced directly from original creation package], or original software format).
 - An editable file of each table (.doc, .docx, .xls, .xlsx, or .csv).
 - An editable file of all figure and table captions.
- Note: you may upload the figure, table, and caption files in a single Zip folder.
- Any electronic supplementary material (ESM).
 - If you are requesting a discretionary waiver for the article processing charge, the waiver form must be included at this step.
 - If you are providing image files for potential cover images, please upload these at this step, and inform the editorial office you have done so. You must hold the copyright to any image provided.
 - A copy of your point-by-point response to referees and Editors. This will expedite the preparation of your proof.

- Ensure that your data access statement meets the requirements at <https://royalsociety.org/journals/authors/author-guidelines/#data>. You should ensure that you cite the dataset in your reference list. If you have deposited data etc in the Dryad repository, please include both the 'For publication' link and 'For review' link at this stage.
- If you are requesting an article processing charge waiver, you must select the relevant waiver option (if requesting a discretionary waiver, the form should have been uploaded at Step 3 'File upload' above).
- If you have uploaded ESM files, please ensure you follow the guidance at <https://royalsociety.org/journals/authors/author-guidelines/#supplementary-material> to include a suitable title and informative caption. An example of appropriate titling and captioning may be found at https://figshare.com/articles/Table_S2_from_Is_there_a_trade-off_between_peak_performance_and_performance_breadth_across_temperatures_for_aerobic_scope_in_teleost_fishes_/3843624.

Author's Response to Decision Letter for (RSOS-211171.R0)

See Appendix A.

Decision letter (RSOS-211171.R1)

Dear Miss Sutton,

On behalf of the Editors, we are pleased to inform you that your Manuscript RSOS-211171.R1 "Quantity over quality? Prey-field characteristics influence the foraging decisions of little penguins (*Eudyptula minor*)."

subject to minor revision in accordance with the referees' reports. Please find the referees' comments along with any feedback from the Editors below my signature.

Please submit your revised manuscript and required files (see below) no later than 7 days from today's (ie 28-Feb-2022) date. Note: the ScholarOne system will 'lock' if submission of the revision is attempted 7 or more days after the deadline. If you do not think you will be able to meet this deadline please contact the editorial office immediately.

on behalf of Dr Agustina Gómez-Laich (Associate Editor) and Pete Smith (Subject Editor)
openscience@royalsociety.org

Associate Editor Comments to Author (Dr Agustina Gómez-Laich):

Dear authors,

Thank you for this new version of the manuscript and the point by point detailed reply to reviewer's comments. The manuscript is clearer now, however I have some additional comments and suggestions that are listed below.

Line 44. "IMASEN" refers to the intra-mandibular angle sensors that can be used to determine when birds open their beak. This technique has been employed on several penguin species in order to estimate prey consumption. I suggest incorporating this information. You can find references here: <https://www.sciencedirect.com/science/article/pii/S1873965207000229>

Line 69. Please incorporate the little penguin's specific name.

Line 80-84. I suggest rephrasing this prediction since it is not clear.

Line 90. Please, add "and" before "weighed in a cloth bag".

Line 97. "were" instead of was?

Line 117. Please include what was defined as a dive bout. Some information is presented below but please specify for example after how much time at the surface a dive bout was considered to have ended.

Line 136-137. Please rephrase. The sentence starts and ends with "after".

Line 142. "Using" instead of "in"

Line 147. Did the model only include prey type? The results mentioned that it included prey type, patch density and the interaction between both.

Line 154. 'mean'

Line 181. Here it says that this was the only model with $\Delta AICc$ of < 4 however what a $\Delta AICc$ of < 4 means is never mentioned in the methodology. Please incorporate this information into the methods section.

Line 182-183. Can you please explain what this interaction means in a clearer way? It is not totally clear what you are trying to say here. If penguins spend more time at patches where prey density is highest, and these patches are of larval fish it is not clear why penguins spend less time at patches of sprat and larval fish than anchovy. These sentences are confusing.

Line 183. "at" is missing here.

Line 196-197. This sentence sounds awkward. Please rephrase it.

Line 212. "Penguins' foraging range

Figure 1 Caption. Schematic "for" ?

Figure 3 caption. Please rephrase this figure caption. The second sentence includes an explanation, this information should be placed on Results not here.

Table 2. Please explain more in detail what each of the columns mean. For example, it is not clear why prey capture events and fewer than total prey captures. What does % of patch events refers to?

===PREPARING YOUR MANUSCRIPT===

one version should clearly identify all the changes that have been made (for instance, in coloured highlight, in bold text, or tracked changes);

===PREPARING YOUR REVISION IN SCHOLARONE===

-- If you are requesting an article processing charge waiver, you must select the relevant waiver option (if requesting a discretionary waiver, the form should have been uploaded, see 'File upload' above).

-- If you have uploaded any electronic supplementary (ESM) files, please ensure you follow the guidance at <https://royalsociety.org/journals/authors/author-guidelines/#supplementary-material> to include a suitable title and informative caption. An example of appropriate titling and captioning may be found at https://figshare.com/articles/Table_S2_from_Is_there_a_trade-

off_between_peak_performance_and_performance_breadth_across_temperatures_for_aerobic_sc_ope_in_teleost_fishes_/3843624.

Author's Response to Decision Letter for (RSOS-211171.R1)

See Appendix B.

Decision letter (RSOS-211171.R2)

Dear Ms Sutton,

I am pleased to inform you that your manuscript entitled "Quantity over quality? Prey-field characteristics influence the foraging decisions of little penguins (*Eudyptula minor*).\" is now accepted for publication in Royal Society Open Science.

Please remember to make any data sets or code libraries 'live' prior to publication, and update any links as needed when you receive a proof to check - for instance, from a private 'for review' URL to a publicly accessible 'for publication' URL. It is also good practice to add data sets, code and other digital materials to your reference list.

Royal Society Open Science is a fully open access journal. A payment may be due before your article is published. Our partner Copyright Clearance Centre will contact the corresponding author about your open access options (if you have any queries regarding fees, please see <https://royalsocietypublishing.org/rsos/charges> or contact authorfees@royalsociety.org).

on behalf of Dr Agustina Gómez-Laich (Associate Editor) and Professor Pete Smith (Subject Editor).

Follow Royal Society Publishing on Twitter: @RSocPublishing
Follow Royal Society Publishing on Facebook:
<https://www.facebook.com/RoyalSocietyPublishing/>
Read Royal Society Publishing's blog:
<https://royalsociety.org/blog/blogsearchpage/?category=Publishing>

Appendix A

Firstly, we would like to express our gratitude to the reviewer and editor for considering this manuscript and providing us longer for the turn around on revisions. We hope you will find the manuscript now to be much improved. We have addressed all the comments and provided line numbers (matching the tracked changes document) where applicable and noting responses with initials.

Reviewer comments

Introduction (and throughout): with respect to the data on time spent in prey patches, the term “foraging success” should be changed to “foraging effort”, since the two are not necessarily equivalent but seem to be used interchangeably here.

GJS: Foraging effort is mainly used throughout the manuscript apart from one instance on line 74. The term foraging success is appropriate here since individuals must maximise their success not their effort.

Line 44. In addition to GPS and time-depth recorders, there are other methods such as the imasen, stomach temperature sensors and accelerometry that allow estimating prey capture events. I suggest incorporating this information.

GJS: I think potentially imasen is a spelling mistake. Nevertheless, I have now included stomach temperature sensors and accelerometry in the manuscript with appropriate references. Please see line 44

Line 52. Please incorporate more references here. For example:

1) Akiyama Y, Akamatsu T, Rasmussen MH, Iversen MR, Iwata T, Goto Y, et al. (2019) Leave or stay? Video-logger revealed foraging efficiency of humpback whales under temporal change in prey density. PLoS ONE 14(2): e0211138.

<https://doi.org/10.1371/journal.pone.0211138>

2) Watanabe YY, Ito M, Takahashi A. Testing optimal foraging theory in a penguin-krill system. Proc Biol Sci. 2014 Jan 29;281(1779):20132376. doi: 10.1098/rspb.2013.2376. PMID: 24478293; PMCID: PMC3924065.

GJS: These references have now been added to the manuscript. Please see line 53

Line 79. It would greatly improve the manuscript if not only the hypothesis but also the derived predictions are presented.

GJS: The predictions are based on the optimal foraging theory and are briefly mentioned in line 78-80. We have now restructured this section to present our predictions after the hypothesis. Please see paragraph beginning line 77

Were all the videos recorded simultaneously, so that all individuals can be assumed to have reasonably encountered the same prey aggregations? Providing some indication of the temporal separation of sampling by individuals will be helpful for considering how changes in prey availability over a season may affect foraging behaviors.

GJS: Videos from each colony were recorded over a 7-10 day period so all individuals at both colonies would have experienced some of the same prey aggregations. However, the question we are investigating is more on the line of what directly at the patch level and not the environment level patch density. Nevertheless, I have indicated the exact sampling periods for both colonies. This information is now on in the deployment summary table 1.

Line 86: clarify what was measured with calipers, please

GJS: Morphometrics bill length, depth, head length and flipper were measured with calipers and ruler. This information is now on line 91.

Line 90. Did cameras record continuously for 15 minutes every hour? I believe they did but please make this clear.

GJS: Yes, I have now included this on line 94

Line 93. Please incorporate penguins' body mass. How much time did the instrumentation procedure last? Were all penguins instrumented in the same period? You said instrumentations were performed between October and January. It would be nice to incorporate a table with the instrumentation date of each animal. How were animals captured? Did you have a control group in order to determine if the presence of a camera and GPS had an effect on penguins' foraging behaviour (i.e. foraging trip duration)? After instrumentation, did birds continue breeding normally? Please incorporate these pieces of information.

GJS: I have now populated this section with the requested information. Please see line 98 for revisions. Birds continued to forage normally since we retrieved all devices and nests were checked after deployment to see if adults were still returning to the burrows. We did not have a control group. However, due to the fact that London bridge colony is a land-based colony, it meant I could perform regular nest checks at 2 week intervals after the deployment period until the chicks fledged. This was not possible for GI and is why this information is not included in the study.

Line 96. You state that all analyses were performed with R. Even video visualization?

GJS: This has now been corrected to include "Unless otherwise stated". Thank you for pointing that out.

Line 101. It is not clear why 10 still images from each dive were selected. Couldn't it be better to relate the number of still images per dive to the duration of the dive? Or were all recorded dives similar in duration?

GJS: Apologies, this should be "Dive bout" not dive. 10 images were chosen as a standard in order to obtain an average patch density over the whole foraging bout in order to get an average of density. Although recorded bouts varied in length, 10 still images and 3 counts per image was decided to avoid excessive counting.

Line104-107: How did you translate fish lengths from the literature to appropriately sized polygons on the frame? This needs clarification. Also, did you count everything within those small polygons, or only clearly recognizable prey species? Did you count only individuals that were completely within the frame, or also those crossing the frame to some degree? Much more clarity on methods and assumptions is required to understand how you estimated areal density here.

GJS: As mentioned in line 119, A rectangle based on body length was used for all counting and the body lengths were transformed into real values based on literature after the counts. I counted everything within those small polygons (line 118) since bait balls are homogenous in prey type. Additional text has now been included on line 118.

Line 106. Were these rectangles placed at random on the image?

GJS: They were placed on random locations over the bait ball. This has now been included in the text at line 120

Line 107: do you know the species of larval fish in the imagery? The text here mentioning the dimensions of the polygons suggests you do. If so, replace “larval fish” with the appropriate species names. Else, please describe how a length for generic larval fishes was estimated.

GJS: The species is not identifiable hence why we used “Clupeiformes” we used the average size of larval stage anchovy and sprat to determine the size of juvenile Clupeiformes.

Line 110-113 – presumably these mean and standard deviation statistics refer to the summaries of the raw density estimates from the 3 replicate counts of fish in the frame on each of 10 images per dive on each fish type (N=?) so that you represent the scaled density as a variable with zero mean and unit variance? If so, are the axes in Figure 2 correct? Please clarify. Also, please clearly define the acronyms used.

GJS: The acronyms have been clarified the values presented are essentially z-scores. A standardised value that describes the value’s relationship to the mean group of values. So for example, if the value is close to 0, then that indicates the data points score is close to the mean, if it is 1, then the values is one standard deviation from the mean.

Line 116-117: how was the determination of patch entry/exit quantified? Were these bounded by clear period of transit or failed foraging dives? Also, please present units for the PRT metric here and throughout

GJS: This information is already provided in the next sentence however I have now structured and added additional information to improve clarity. Please see line 132.

Line 117. How did you define when a penguin was entering and leaving a patch? Please clarify this aspect.

GJS: This has been answered with the additional information provided from line 132

Lines 117-119- this assumption may be appropriate if prey depth distributions (and time spent on ascent/descent) is similar across prey patches. If not, is the PRT metric contaminated by variation in behaviors not directly related to foraging? Why not quantify PRT on a per-dive basis that includes only the time within the actual prey patch?

GJS: PRT could not be calculated in a per dive basis as the periods in the patch as short in duration and dive by dive has a low prey capture rate. Although the reviewer point is valid, the paper is assessing geospatial PRT not dive by dive. We decided to do this due to the above mentioned reasons. Little penguins are shallow divers in comparison to other penguin species and time spent on ascent/descent would be similar. As penguins are limited by breath hold, the effort/time in ascent and descent should still be considered in PRT metric as the whole period they are working/using energy.

Line 119. How did you do when the bird entered a patch but the video recording stopped before it left? This is briefly explained in figure 1. Please incorporate this information into the methods section.

GJS: This information is now on line 139

Line 120. PRT instead of (PRT)

GJS: Brackets have been removed

Stats. Which library did you employ to perform the models, and which library did you use for model selection? Please incorporate this information.

GJS: This information is now on line 141 and line 146.

It would be interesting to incorporate some depth data in order to see if decisions are influenced by this variable. Did you have depth data? Could you estimate it by using the time it took little penguins to reach the bottom phase of a dive?

GJS: We did not instrument individuals with a depth recorder as we were already towards the upper limit of device size. Unfortunately, without knowing the angle or speed of a dive is not possible to estimate depth.

It will help to understand the efficiency of the methods if you can provide additional, general summaries of the data used in the study. For example, from the 22 deployments, how many 15min video segments were used and what percent of the total recorded time is included in PRT? Of the video used, did all the individuals exhibit foraging in different prey types or were some specialized on specific prey only? Consider adding some indication of predator foraging success (e.g, mean fish eaten per dive could be added to Table 1, rather than the S1 figure)

GJS: As suggested this information is provided in the manuscript through tables. We did not discuss specialised diet since we are constrained by 15 min intervals.

Line 147-154. In relationship to the previous comments, do penguins capture the different prey types at similar depths? Couldn't it be that anchovy and sprat shoals are more easily broken than larval fish shoals so penguins stay in the larval fish shoals for longer? Authors discuss this briefly, I suggest discussing more in detail how differences in prey behaviour may affect predators decisions.

GJS: As mentioned, it is not possible to estimate depth. As PRT is an estimate of the whole period at the prey patch, not just the period present in the fish shoals. Without depth recorders, it is not possible to assess these in detail.

Reviewer comment: I suggest discussing more the results in the context that what authors report does not consider the energy expenditure associated with capturing different prey types. High caloric prey may not be so profitable if the energy expenditure associated to capture it is high. Not having estimations of the energy expenditure associated to capturing different prey types is a key limitation of the study and something that could be measured in the future for example by means of accelerometry.

GJS: This has been taken into consideration in edits of the discussion, please see lines e.g. line 246 for amendments.

Line 170-172: The authors state the birds showed a reduction in distance and duration. Of what?

Line 170-172. A reduction in distance and time in relation to the non-foraging period? Is this what you are trying to say? Please rephrase the sentence.

GJS: Yes, thank you for pointing that out. I have now amended and discussed in more detail from line 203

Line 184-186- or because it takes longer to fill up on small prey items?

GJS: This is true, although if the patch disassociates then the penguin will not stay. This is already discussed as an alternative in lines 218. I have restructured to provide further information as was as alternative hypotheses please see paragraphs beginning 225 and 239.

Line 202. Eliminate "his"

GJS: This has been removed from the manuscript

Appendix B

We thank the reviewers and editor for provisionally accepting this manuscript. Please see responses to your comments. We hope that the minor editions to the manuscript have now cleared up any ambiguity.

Line 44. "IMASEN" refers to the intra-mandibular angle sensors that can be used to determine when birds open their beak. This technique has been employed on several penguin species in order to estimate prey consumption. I suggest incorporating this information. You can find references here: <https://www.sciencedirect.com/science/article/pii/S1873965207000229>

GJS: Thank you for clarifying. I have now included this in the manuscript on line 44

Line 69. Please incorporate the little penguin's specific name.

GJS: This is now included on line 69

Line 80-84. I suggest rephrasing this prediction since it is not clear.

GJS: I have now restructured this sentence to improve readability.

Line 90. Please, add "and" before "weighed in a cloth bag".

GJS: This has been added on line 90.

Line 97. "were" instead of was?

GJS: This has now been corrected. Please see line 95.

Line 117. Please include what was defined as a dive bout. Some information is presented below but please specify for example after how much time at the surface a dive bout was considered to have ended.

GJS: I have rephrased dive bout to foraging bout as dives can be in sequence but PRT was considered from first dive encountering prey to the end of the final dive. This information is already on line 138. I have included the requested information on line 138 as it is better placed here than on line 177.

Line 136-137. Please rephrase. The sentence starts and ends with "after".

GJS: This has now been removed

Line 142. "Using" instead of "in"

GJS: This has now been changed. Please see line 143

Line 147. Did the model only include prey type? The results mentioned that it included prey type, patch density and the interaction between both.

GJS: Yes it included the interaction too. This has now been rewritten to improve clarity. Please see line 148

Line 154. "mean"

GJS: This has been fixed. It is now on line 155.

Line 181. Here it says that this was the only model with $\Delta AICc$ of < 4 however what a $\Delta AICc$ of < 4 means is never mentioned in the methodology. Please incorporate this information into the methods section.

GJS: This information has now been incorporated on lines 154-55 in the methods section

Line 182-183. Can you please explain what this interaction means in a clearer way? It is not totally clear what you are trying to say here. If penguins spend more time at patches where prey density is highest, and these patches are of larval fish it is not clear why penguins spend less time at patches of sprat and larval fish than anchovy. These sentences are confusing.

GJS: I have restructured this section to improve clarity. Please see paragraph beginning at 181. Basically the story is that on numerical abundances, the story matches the hypothesis (penguins spending longer at anchovy than larval fish). But when considering calorific value of a prey patch, individuals are spending too long at larval fish patches – in fact much longer than at anchovy patches. This is particularly so evident when the patch of larval fish is big.

Line 183. “at” is missing here.

GJS: This section has been changed with the edits above.

Line 196-197. This sentence sounds awkward. Please rephrase it.

GJS: This has now been rephrased and is on line 197.

Line 212. “Penguins’ foraging range

GJS: This has been changed now

Figure 1 Caption. Schematic “for” ?

GJS: This has now been changed

Figure 3 caption. Please rephrase this figure caption. The second sentence includes an explanation, this information should be placed on Results not here.

GJS: This has now been removed

Table 2. Please explain more in detail what each of the columns mean. For example, it is not clear why prey capture events and fewer than total prey captures. What does % of patch events refers to?

GJS: I have now rephrased this to explain the columns in detail.